# Simultaneous SARS-CoV-2 Genome Sequencing of 384 Samples on an Illumina MiSeq Instrument through Protocol Optimization

**DOI:** 10.3390/genes13091648

**Published:** 2022-09-14

**Authors:** Nasserdine Papa Mze, Mamadou Beye, Idir Kacel, Raphael Tola, Leonardo Basco, Hervé Bogreau, Philippe Colson, Pierre-Edouard Fournier

**Affiliations:** 1UMR VITROME, Aix-Marseille University, IRD, AP-HM, SSA, IHU—Méditerranée Infection, 13005 Marseille, France; 2IHU—Méditerranée Infection, 13005 Marseille, France; 3Unité de Parasitologie et Entomologie, Département de Microbiologie et Maladies Infectieuses, Institut de Recherche Biomédicale des Armées, 13005 Marseille, France

**Keywords:** Illumina MiSeq, SARS-CoV-2, next-generation sequencing, genomics, sequence analysis

## Abstract

In the present study, we propose a high-throughput sequencing protocol using aNextera XT Library DNA kit on an Illumina MiSeq instrument. We made major modifications to this library preparation in order to multiplex 384 samples in a single Illumina flow cell. To validate our protocol, we compared the sequences obtained with the modified Illumina protocol to those obtained with the GridION Nanopore protocol. For the modified Illumina protocol, our results showed that 94.9% (357/376) of the sequences were interpretable, with a viral genome coverage between 50.5% and 99.9% and an average depth of 421×. For the GridION Nanopore protocol, 94.6% (356/376) of the sequences were interpretable, with a viral genome coverage between 7.0% and 98.6% and an average depth of 2123×. The modified Illumina protocol allows for gaining EUR 4744 and returning results of 384 samples in 53.5 h versus four times 55.5 h with the standard Illumina protocol. Our modified MiSeq protocol yields similar genome sequence data as the GridION Nanopore protocol and has the advantage of being able to handle four times more samples simultaneously and hence is much less expensive.

## 1. Introduction

SARS-CoV-2, the pathogenic agent that causes Coronavirus disease 2019 (COVID-19), has spread very rapidly around the world, affecting millions of people, and is still circulating with a substantial global incidence [1]. The emergence of this new virus called for the urgent establishment of a global system of reliable diagnosis and close surveillance, including the determination of the genetic epidemiology of SARS-CoV-2. In this context, rapid and efficient methods to simultaneously analyze large numbers of samples are needed for virus whole genome sequencing (WGS). The next generation of sequencing (NGS) technologies, such as those of Oxford Nanopore Technologies (Oxford Nanopore Technologies, Oxford, UK) and Illumina (Illumina, Evry, France), are currently used in most laboratories worldwide [2]. In addition, the ARTIC network developed a primer set (named ARTIC) for performing amplification by PCR that raises the yield of subsequent sequencing of the whole SARS-CoV-2 genome [3]. Due to the rapid emergence of viral variants harbouring multiple new mutations, the design of new ARTIC primers is constantly required; the version currently (in May 2022) in use is ARTIC V4.1, consisting of Pool A, which contains 50 primers, and Pool B, which contains 49 primers [4]. The Minion or GridION Nanopore and MiSeq Illumina platform techniques are often used with ARTIC primers [5,6] to sequence up to 96 samples in a single flow cell.

Recently, a new protocol for the next-generation sequencing of SARS-CoV-2 RNA using the Illumina MiSeq instrument was described [7]. Minor modifications were introduced to the Illumina COVIDSeq protocol to sequence 94 SARS-CoV-2-positive samples on a MiSeq instrument. The COVIDSeq protocol is most often used to sequence a much larger number of samples (i.e., from 384 to 3072 samples) on a NovaSeq instrument. However, this instrument is very expensive (up to one million euros per run, equivalent to the price of a NovaSeq 6000 sequencing platform), especially for developing countries. It is in this context that we propose a simple, rapid, and less expensive method to sequence 384 SARS-CoV-2 amplicons in a single run on a MiSeq instrument. To validate our protocol, we compared the results obtained using this method with the Illumina MiSeq instrument to those obtained with the GridION Nanopore instrument using four flow cells and the ARTIC Nanopore protocol Eco PCR tilling of SARS-CoV-2 with Native Barcoding.

## 2. Materials and Methods

### 2.1. Samples

The study was conducted using RNA extracts obtained from nasopharyngeal swabs collected from patients for the purpose of SARS-CoV-2 clinical routine diagnosis by real-time reverse transcription-PCR (qPCR) and found SARS-CoV-2 RNA-positive. These RNA extracts were then used for SARS-CoV-2 genome sequencing for the purpose of genomic surveillance following recommendations of the French public health authorities (https://www.santepubliquefrance.fr/dossiers/coronavirus-covid-19/consortium-emergen#block-356295, accessed on 13 September 2022). This study received approval of the Ethical Committee of the Mediterranee Infection Institute under reference (No. 2022-017). Confidentiality and anonymity were maintained by using a code number on individual clinical records. Only samples from patients who gave their written informed consent were used in the study.

### 2.2. RNA Extraction

Viral RNA had been extracted from nasopharyngeal swab samples using an automated nucleic acid purification system (Thermo Scientific KingFisher Flex Purification System, Waltham, MA, USA) following the MVP_2Wash_200_Flex protocol as recommended by the manufacturer. After extraction, eluted RNA is stored at −20 °C until further processing.

### 2.3. Modified Illumina MiSeq Protocol

#### 2.3.1. cDNA Synthesis

The first strand of cDNA was synthesized using a LunaScript RT SuperMix kit (New England Biolabs, Ipswich, MA, USA). Two µL of LunaScript were mixed with 8 µL of eluted RNA, and the mixture was incubated in the thermal cycler using the following program: 25 °C for 2 min, 55 °C for 10 min, 95 °C for 1 min and hold at 4 °C.

#### 2.3.2. Amplification of cDNA

The synthesized cDNA was amplified in two separate PCR reactions: PCR Pool A and PCR Pool B (Integrated DNA Technologies, Leuven, Belgium). Two separate mixes were prepared for each sample, one for Pool A (for primer set A) and the other for Pool B (for primer set B) [4], consisting of 0.37 µL of each primer (100 µM), 9.63 µL of nuclease-free water, and 12.5 µL of master mix (Q5 hot start high-fidelity 2× master mix; New England Biolabs), into which we added 2.5 µL of the cDNA to each pool mix well. PCR amplification was carried out using the following program: initial denaturation for 30 s at 98 °C, followed by 35 cycles of denaturation for 30 s at 98 °C, and 5 min of annealing and extension at 65 °C. The PCR products of each sample (25 µL of Pool A and 25 µL of Pool B) were mixed (Pool A/PoolB) and diluted 1:10 in water as recommended in the protocol for Eco PCR tiling of SARS-CoV-2 virus with native barcoding [8].

#### 2.3.3. Tagmentation of Genomic DNA

This step used the Nextera XT DNA Library kit (Illumina, catalogue number: FC-131-1096). Five µL of each sample were mixed with 10 µL of Tagment DNA buffer and 5 µL of Amplicon Tagment Mix and incubated at 55 °C for 5 min with a hold at 10 °C. At 10 °C, 5 µL of NT were immediately added to stop tagmentation prior to incubation for 5 min at room temperature.

#### 2.3.4. Amplification, Clean up and Pooling of Libraries

This step amplified the tagmented DNA using 10 μL of index adapters (Integrated DNA Technologies(IDT), Coralville, IA, USA)for Illumina PCR index sets 1–4 (Illumina, catalogue number: 20043137)mixed with 15 µL of Nextera PCR Master Mix (Illumina). PCR was carried out using the following conditions: initialization for 3 min at 72 °C, initial denaturation for 3 min at 98 °C, followed by 12 cycles of denaturation for 20 s at 98 °C, annealing for 5 min at 65 °C, extension for 2 min at 72 °C, and a final extension step for 5 min at 72 °C. Five µL of each sample were pooled to obtain 440 μL of pooled libraries (for 96 samples), and a total of 4 pools of samples (4 × 96 = 384) were constituted. Each library pool was purified with 396 μL of Illumina tune beads (0.9 beads ratio) and washed twice with 1000 µL of 80% ethanol freshly prepared. Elution was performed using 55 µL of resuspension buffer of the Illumina kit. Twenty-five µL of each pool were transferred to a tube to form a mega pool and were diluted to 8 nM with Resuspension Buffer.

#### 2.3.5. Sequencing of the Libraries

During this step, DNA samples were denatured by adding 10 µL of 0.2 N NaOH mixed with 10 µL of mega pool (8 nM) and incubated for 5 min. A total of 980 µL of neutralizing buffer (HT1) was added to obtain the final volume of 1000 µL in the library. A total of 180 µL of the final library was mixed with 420 µL of HT1, and 6 µL of this mixture was removed and replaced with 6 µL of 20 pMphix. The library was loaded into the MiSeqsequencing cartridge with a concentration of 24 pM to obtain a density of 1000 K/mm^2^ without overclustering the flow cell (MiSeq Reagent Kit V2, catalogue number: MS-102-2003). Sequencing was initiated as per the MiSeq sequencing guide by Illumina.

### 2.4. Nanopore Protocol

We used the same amplicons prepared and diluted to 1:10, as described above in Section 2.3.2, to sequence them with the GridION system (Oxford Nanopore Technologies), following the protocol “Eco PCR tiling of SARS-CoV-2 virus with native barcoding (EXP-NBD104, EXP-NBD114, and EXP-NBD196) Version: PTCE_9122_v109_revC_10Feb2021” [8]. cDNA was treated with the NEBNext Ultra II End repair/dA-tailing Module (New England Biolabs). It was ligated with barcodes from a native barcoding kit (Oxford Nanopore Technologies) using Blunt/TA Ligase Master Mix (New England Biolabs). The adapters from the ligation sequencing kit (Oxford Nanopore Technologies) were then ligated with the NEBNext Quick Ligation Module (New England Biolabs). Finally, sequencing was performed using the GridION sequencer with four flow cells (R9.4.1) for 48 h.

### 2.5. Genome Assembly, Variant Calling, and Phylogenetic Analysis

These steps describe the basic bioinformatics pipeline for SARS-CoV-2 genome assembly and identification of variants. The raw sequencing data generated by Illumina sequencing platforms were in the form of FASTQ format before further processing. The raw data generated by Nanopore sequencing platforms were in the form of FASTQPASS format.

### 2.6. Illumina Sequencing Data Analysis

Quality control and adaptor trimming of raw reads were performed using trimmomatic(v. 0.39) [9]. The trimmed reads were aligned against the SARS-CoV-2 reference genome using minimap2 (v2.17-r941) [10]. With samtools(v. 1.13), primers have been removed, and the sam files from the mapping were sorted and converted to bam [11]. Using sam2consensus [12], a consensus sequence of the genomes in fasta file format was generated. To assess the quality of the assembled genomes, sample parameters and metrics are shown in Appendix A.

### 2.7. Nanopore Sequencing Data Analysis

Reads were collected and filtered using the artic-ncov2019 bioinformatics pipeline [13]. The reads were filtered with a minimum length of 400 base pairs (bp) and a maximum length of 700 bp using guppyplex. The consensus sequence in the fasta file was generated using the ARTIC pipeline with the Medaka model [14].

### 2.8. Consensus Sequences Analysis

The consensus sequences in fasta format were analyzed using Nextclade (https://clades.nextstrain.org/, accessed on 13 September 2022) [15] to determine the Nextstrain clades and identify mutations (https://nextstrain.org/, accessed on 13 September 2022 [15]. To assign lineages to the assembled genome fasta sequences, we use the Phylogenetic Assignment of Named Global Outbreak Lineages (PANGOLIN) software [16]. The comparison of data output, coverage, and time taken for analysis between MiSeq and Nanopore procedures is presented in Appendix A.

## 3. Results

Of the 376 samples processed for sequencing with the modified Illumina MiSeq protocol, 357 (94.9%) yielded interpretable sequences, as defined by nexclade. Of these, 325 had high quality sequences with at least 90% coverage of the SARS-CoV-2 genome and a mean depth (±standard deviation) of 407 ± 248.92 × (range, 158.50–656.35). Thirty-two samples had coverage between 50.6% and 89.9% and a mean (±standard deviation) depth of 561 ± 305.93×(255.46–867.3) (Appendix A). Sequencing of 19 samples failed, and of these, 13 had medium or low viral loads, with real-time cycle threshold (C_T_)values ranging from 23.0 to 33.7 (Figure 1). Six of 19 samples produced no reads for unknown reasons. The MiSeq genome sequences with a high quality (*n* = 351) were submitted to GenBank (accession numbers ON365966–ON366316). In all samples sequenced with the modified MiSeq protocol, an average of 59.3 mutations was found. In total, 19 different SARS-CoV-2 lineages were detected in the 357 samples that produced sequences with the modified MiSeq workflow (Figure 2B).

For the same samples processed by Nanopore GridION sequencing, 356 (94.6%) yielded interpretable sequences. All 356 samples with interpretable data were also sequenced successfully with the modified MiSeq protocol. Among 356 Nanopore genome sequences, 332 were of very good quality, with a coverage of more than 90% and a mean depth of 2238 ± 892.41 × (1345.70–3130.54) (Appendix A). Twenty-three samples had a genome coverage between 50.78 and 89.21% and a mean depth of 530.25 ± 238.30 × (291.94–768.55). All 19 samples that failed to yield interpretable genome sequences with the modified MiSeq protocol also failed to yield interpretable sequences with Nanopore.

Two samples (IHUCOVID-071236 and IHUCOVID-071258) yielded interpretable sequences with the modified Illumina MiSeq protocol although they could not be sequenced by the Nanopore system. Genome coverages for these samples were 7.0% and 99.6%, and sequencing depths were 931× and 273×, respectively. Conversely, one sample (IHUCOVID-071471) yielded an interpretable sequence with the Nanopore protocol but not with the modified MiSeq protocol. Genome coverage was 58.6%, and sequencing depth was 2965×. The reason for these discrepancies is unresolved but might be handling errors. The Nanopore genome sequences with a high quality (*n* = 327) were submitted to GenBank (accession numbers ON313815–ON314140). On average, the number of mutations found with Nanopore system was 58.2.

For samples from which SARS-CoV-2 genomes were sequenced with both methods, Pangolin assigned the same lineage in 355 of 356 (99.7%) cases (Appendix A). For one sample, the MiSeq consensus genome was assigned to the B.1.177 variant, while the Nanopore consensus genome was assigned to the B.1.177.31 lineage of this variant (Figure 2A). Genome depth was greater with the Nanopore protocol (mean, 2023.8 ± 1047.96) compared to the modified protocol (422.8 ± 266.50), and the difference was significant (*p* = 0.0001) using the *t*-test to compare these two means. The coverage was similar between the two techniques (96.57 ± 7.2% on average for the Nanopore protocol versus 95.6 ± 5.62% on average for the modified protocol), but there was a significant difference (*p* = 0.04). However, there were more mutations detected (59.3 versus 58.2 on average) with the modified MiSeq protocol but no significant difference (*p* = 0.9591)were detected using the *t*-test.

Furthermore, to sequence 384 samples, the total cost was estimated to be EUR 15,629 using the modified Illumina protocol, versus EUR 20,374 for the standard Illumina protocol (Figure 3). The cost saving was estimated to be EUR4745 using the modified Illumina protocol on a single MiSeq sequencer compared to the standard Illumina protocol that has to be performed on four MiSeq sequencers. Therefore, the modified protocol saves money and gives results of 384 samples in 53.5 h while sparing the availability of three MiSeq instruments.

## 4. Discussion

The SARS-CoV-2 genome is rapidly evolving. Close monitoring of their evolution is therefore necessary for a better assessment and understanding of potential change in transmissibility, pathogenicity, and immune evasion. Several NGS methods are used for whole SARS-CoV-2 genome sequencing. However, existing techniques may be improved in terms of rapidity and cost in the face of the pandemic. Moreover, currently used reagents are expensive and this can notably hamper genomic surveillance in developing countries. Therefore, it is important to tune in simple, rapid, high-throughput, and less expensive methods implementable in all countries affected by the pandemic.

The Illumina platform is known for its capacity to handle a large number of samples simultaneously by multiplexing samples and for its high yield of sequences [17]. The MiSeq sequencing technique is usually used to sequence 96 samples in a single flow cell, depending on the different libraries used [7,18]. In the present study, we optimized the protocol to sequence 384 samples in a single flow cell. The results obtained with the modified Illumina protocol showed high quality sequences despite the increase in the number of samples to be sequenced. However, using the Nanopore protocol, a greater sequencing depth was obtained compared to the modified Illumina protocol (Figure 1). This difference is probably due to the large number of samples used with the modified Illumina protocol, compared to the Nanopore protocol. Indeed, the sequencing depth depends on the amount of reads generated, which varies according to the NGS protocol. As the number of samples increases, the sequencing depth decreases. Nonetheless, the sequence data derived from both protocols were identical, with the exception of a single sample.

Among the samples analyzed in this study, 19 lineages were characterized (Figure 2B), including the variants 21K, 21L, and 21M (B.1.1.529 for two samples) that circulated recently in Europe and required monitoring, as recommended by the World Health Organization [19]. In this study, sequencing failed in 19 samples. Thus, genome coverage was low (<50%), and no lineage could be identified after analysis. In order to optimize the NGS protocols assessed in our study, the number of PCR cycles could be increased to 40 for all samples with C_T_ values > 21. In addition, a dilution for samples with C_T_ values > 21 should be avoided. Prior NGS, increasing the amount of respiratory sample used for RNA extraction and reducing the eluted volume may also help optimize the output [20].

One of the main advantages of the modified Illumina protocol is that one laboratory technician can generate all the libraries with minimal handling time, unlike for the ARTIC Nanopore protocol and the standard Illumina protocol. The working time and reagent cost of library preparation are estimated to be 4 h and EUR 12,767, respectively, for the modified Illumina protocol with a single person handling all the samples. By contrast, the same workload would require about 6 h and EUR 15,261 for the standard Illumina protocol, with four laboratory technicians to handle each plate with 96 samples. The difference in price can allow significant cost savings, especially in developing countries. For example, a single MiSeq flow cell costs about EUR 1023 when used for the modified Illumina protocol, compared to 4 flow cells that cost EUR 4092 in total for sequencing 384 samples using the standard Illumina protocol (Figure 3). The cost factor should also take into consideration technical worker requirements as the modified Illumina protocol can potentially reduce the workforce by four-fold for the same quantity of samples processed. Accordingly, the cost of technical staff will decrease by EUR 800. In addition, NGS sequencing procedures require high-level training and skills, which can become an obstacle in laboratories in developed and developing countries. In our experience, 44 h (run sequencer time) was required to obtain the sequences of 384 samples using the modified Illumina protocol, compared to 24–48h with the Nanopore GridION protocol, depending on the C_T_ values of the samples.

## 5. Conclusions

The modified Illumina MiSeq ARTIC protocol allowed us for the first time to sequence a high number (i.e., 384) of SARS-CoV-2-positive samples in a single Illumina MiSeq flow cell. We obtained high quality sequences that gave the same lineage results as the GridION Nanopore platform, at a lower cost.

## Figures and Tables

**Figure 1 genes-13-01648-f001:**
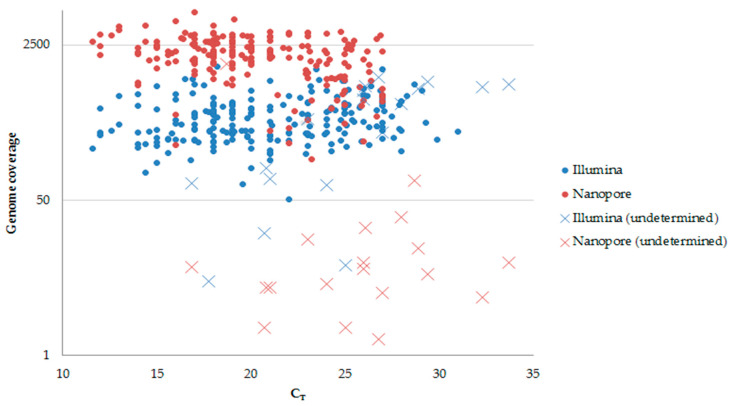
Representation of the genomic coverage according to the C_T_ values.

**Figure 2 genes-13-01648-f002:**
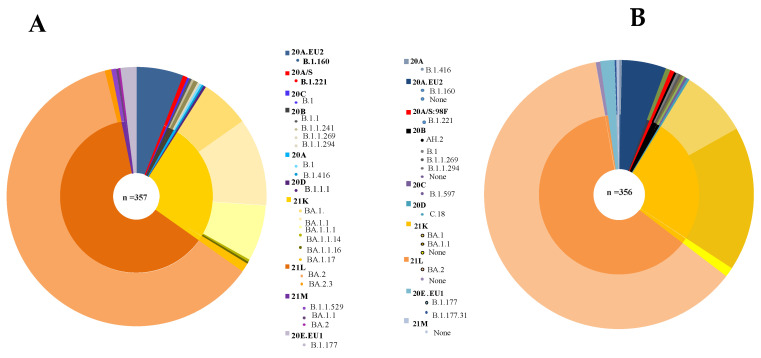
Phylogenetic distribution of SARS-CoV-2 genomes. (**A**) The 357 sequences obtained with the Illumina MiSeq protocol reported in this study are highlighted. The genomes analyzed in this study fall into one of 10 clades, with 21L being the dominant clade. (**B**) The 356 sequences obtained with the Nanopore GridION protocol reported in this study are highlighted. Lineage 21L is still the dominant one. Some PANGOLIN lineages, such as BA.1.1, BA.1.1.1.14, BA.1.1.16, BA.1.17, and BA.1.177, have not been generated.

**Figure 3 genes-13-01648-f003:**
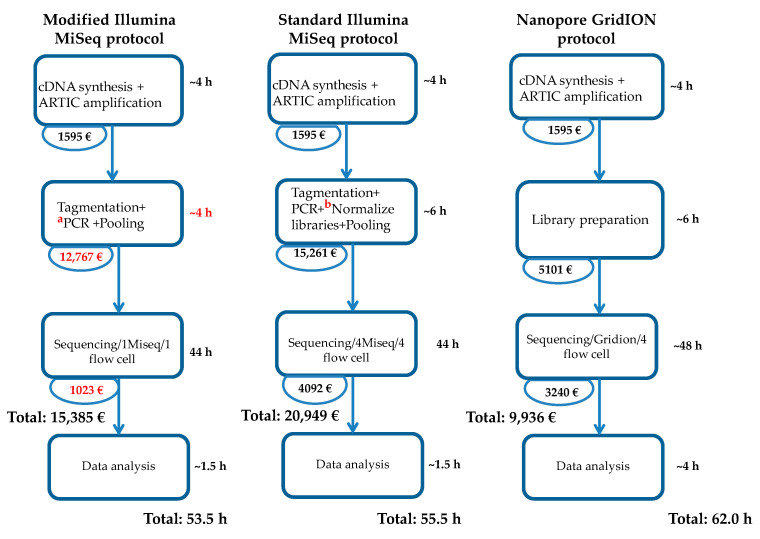
Workflow for each of the described methods, and cost and preparation time for each step. ^a^ During this PCR, the IDT indexes were used, which are less expensive (EUR 1267) and easy to use, contrary to the Nextera XT indexes (Nextera XT Index Kit v2 Set A, Set B, Set C, Set D; catalog number: FC-131-2001) of the standard Illumina protocol, which cost EUR 3517. ^b^ This library normalization step takes approximately 1 h 20 min and is done using the following reagents: Library Normalization Additives 1, Library Normalization Beads 1, Library Normalization Wash 1, Library Normalization Storage Buffer 1 and NaOH (see Nextra XT DNA Library Prep protocol, Document #15031942 v05). However, this normalization step is eliminated using the modified Illumina MiSeq protocol. Only Hybridization Buffer was used for simple normalization as specified in the methodology of this paper.

## Data Availability

The high-quality genomic sequences were submitted to the GenBank (accession numbers ON365966–ON366316 for the modified Illumina MiSeq protocol and ON313815–ON314140 for the Nanopore GridION protocol).

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
