# Peer review of "Simultaneous SARS-CoV-2 Genome Sequencing of 384 Samples on an Illumina MiSeq Instrument through Protocol Optimization"

_genes, 2022, doi:10.3390/genes13091648_

Round 1
Reviewer 1 Report
In this manuscript, Papa Mze et al presented the results of sequencing 384 SARA-Cov-2 samples in a single Illumina MiSeq flow cell via improving the protocol. This improved method can reduce sequencing cost, shorten data turnaround time, and importantly, the quality of the sequencing data is comparable to that of data generated on the GridION Nanopore platform. thus it is of great interest to many field workers using high-throughput sequencing to monitor COVID-19 pandemic and study the evolution of the virus. Overall, the manuscript is well written. But a couple issues need to be addressed.
Major issue
1. The authors gave a detailed description of the optimized protocol for higher multiplex sequencing. But it is not clear which steps are optimized relative to the previous standard protocol, besides using more indices for higher multiplexing. Please clarify in Figure 3 or its legend, or in the Methods Section.
Minor issues
2. Line 16, “94.9% (376/357)” should be “94.9% (357/376)”. It is more readable if changing “coverage” to “viral genome coverage”. Line 17, “94.6% (376/376)”? 100%?
3. Lines 80-90, what are the primer sets A and B?
4. Lines 118-119, I suppose the amplicon used for Nanopore sequencing is the product from Methods 2.3.2, right? Please explicitly specify it.
5. Line 136, did the authors performed quality trimming, adaptor trimming, or both?
6. Lines 149-150, here the authors only mentioned that PANGOLIN was used for phylogenetic analysis of the assembled viral sequences. But based on the Results, the same analysis was performed on the assembled Miseq data.
7. Figures 1A and 1B are largely redundant. To be concise, a combination of two plotting symbols (sequencing platforms) and two colors (success or failure to return sequencing data) can clearly present the data.
8. Lines 163 and 185, how were those mutations identified? Please describe this in the Methods section.
9. Line 174, this statement refers to those two samples mentioned in lines 171-173? Please make it clear. Similarly for Lines 176-177.
10. Lines 171-177, any explanation to these platform-specific failure? Please discuss this a little bit.
11. Line 208, “one Miseq compared to … four Miseq” is not accurate. Do the authors mean one (four) Miseq sequencer(s)?
Reviewer 2 Report
Simultaneous SARS-CoV-2 genome sequencing of 384 samples 2 on an Illumina MiSeq instrument is well written.
Many minor corrections are needed which are mentioned in the PDF as comments.
